# Clinical Characterization and Prognostic Value of TPM4 and Its Correlation with Epithelial–Mesenchymal Transition in Glioma

**DOI:** 10.3390/brainsci12091120

**Published:** 2022-08-24

**Authors:** Jin Wang, Ying Yang, Bo Du

**Affiliations:** 1Department of Emergency, Shenzhen People’s Hospital (The Second Clinical Medical College, Jinan University, The First Affiliated Hospital, Southern University), Shenzhen 518020, China; 2Department of Pediatrics, Futian Women and Children Health Institute, Shenzhen 518045, China

**Keywords:** tropomyosin 4, TPM4, glioma, epithelial–mesenchymal transition, prognosis

## Abstract

Tropomyosin 4 (TPM4) has been reported as an oncogenic gene across different malignancies. However, the role of TPM4 in glioma remains unclear. This study aimed to determine the clinical characterization and prognostic value of TPM4 in gliomas. Transcriptome expression and clinical information were collected from the CGGA and TCGA datasets, which included 998 glioma patients. ScRNA-seq data were obtained from CGGA. R software was utilized for statistical analyses. There was a positive correlation between TPM4 and WHO grades. IDH-wildtype and mesenchymal subtype gliomas were accompanied by TPM4 upregulation. GO and GSEA analysis suggested that TPM4 was profoundly associated with epithelial-to-mesenchymal transition (EMT). Subsequent GSVA revealed a robust correlation between TPM4 and three signaling pathways of EMT (hypoxia, TGF-β, PI3K/AKT). Furthermore, TPM4 showed a synergistic effect with mesenchymal biomarkers, particularly with N-cadherin, Slug, Snail, TWIST1, and vimentin. ScRNA-seq analysis suggested that higher TPM4 was mainly attributed to tumor cells and macrophages and associated with tumor cell progression and macrophage polarization. Finally, high TPM4 was significantly associated with unfavorable outcomes. In conclusion, our findings indicate that TPM4 is significantly correlated with more malignant characteristics of gliomas, potentially through involvement in EMT. TPM4 could predict worse survival for patients with glioma.

## 1. Introduction

Gliomas account for more than 70% of primary brain tumors in adults [1]. Despite advanced diagnostic and therapeutic strategies developed for cancer, frustratingly, gliomas are almost incurable. Especially for patients who suffer from the most malignant type of gliomas, glioblastoma (GBM), the median survival time remains less than fifteen months [2]. Identification of new molecular targets may bring us new opportunities to conquer this fatal disease.

Tropomyosin 4 (TPM4), a member of the tropomyosin family, has been widely reported across different malignancies, including oral squamous cell carcinoma (OSCC) [3], breast cancer [4,5], lung cancer [6,7], hepatocellular carcinoma (HCC) [8], ovarian cancer [9], uterine cervix cancer [10], and colon cancer [11]. Tropomyosin is an actin-binding protein that is supposed to be essential for the contraction of the muscle cell and for maintaining the cytoskeleton of non-muscle cells [12]. The cytoskeleton consisting of the tropomyosin–actin complex is a key regulator of cell morphology, which plays a vital role in cell adhesion and migration in particular pathophysiological processes, such as tumorigenicity, coagulation, and immune response [12,13,14]. Multiple studies tried to investigate the relationship between TPM4 expression and clinical characterization across different tumors but yielded inconsistent results. Higher TPM4 was reported to be associated with more malignant characteristics in most cancers, including oral squamous cell carcinoma [3], breast cancer [4,5], lung cancer [6,7], and hepatocellular carcinoma [8] as well as ovarian cancer [9]. On the contrary, in uterine cervix cancer [10] and colon cancer [11], higher TPM4 predicted a better prognosis.

Currently, there is a lack of knowledge on the role of TPM4 in glioma. Therefore, we focused on TPM4 to search for a new biomarker of glioma. In this study, a total of 998 patients with transcriptional data and clinical information were comprehensively analyzed, aiming to elucidate the clinical characterization and the potential biological mechanisms of TPM4 in glioma development.

## 2. Materials and Methods

### 2.1. Data Collection

In this article, two large public datasets, including The Cancer Genome Atlas (TCGA) [15] dataset and the Chinese Glioma Genome Atlas (CGGA) [16,17,18] dataset, were used for analysis. Transcriptional data and clinical information of glioma patients were downloaded from the corresponding websites. A total of 998 patients with glioma, including 697 samples from the TCGA RNA-sequencing data (level 3, RSEM-normalized) and 301 patients from CGGA301 micro-array data (normalized by GeneSpring GX 11.0), were collected and analyzed. Appendix A describes the baseline clinical characteristics of both cohorts. In addition, the single-cell RNA-seq data (sc-RNAseq) of glioma patients were obtained from CGGA, which consisted of 6148 cells collected from 73 regions of 14 patients. WHO grade information of each patient was also available in the CGGA sc-RNAseq dataset.

### 2.2. Data Preprocessing

For the TCGA dataset, RNAseq data (RSEM-normalized) were converted with log2 transformation. For the CGGA dataset, microarray data were well-preprocessed (normalized, centered, and scaled) by the CGGA database. According to WHO 2021 classification scheme for central nervous system tumors, grade IV gliomas with IDH-mutant can no longer be regarded as GBMs. Thus, GBM cohorts in this study were defined as those only with IDH-wildtype. During the survival analysis, patients with overall survival (OS) of less than 30 days and those without OS information were excluded from the Kaplan–Meier curve and Cox regression analysis. For the sc-RNAseq data, CGGA had already excluded the low-quality genes and low-quality cells. The percentage of mitochondria-expressed genes was less than 5% in CGGA.

### 2.3. TPM4 Highly Correlated Genes and Gene Ontology (GO) Analysis

Pearson correlation tests between all genes and TPM4 were performed to identify TPM4 highly correlated genes. Genes with |correlation coefficients| > 0.6 and *p* value < 0.05 were defined as TPM4 highly correlated genes, which were subsequently used for GO analysis with the online method (DAVID website, https://david.ncifcrf.gov/, accessed on 1 May 2022).

### 2.4. Gene Set Enrichment Analysis (GSEA)

The gene sets used for GSEA analysis were provided in h.all.v7.5.1.symbols.gmt, which was obtained from the MSigDB database (https://www.gsea-msigdb.org, accessed on 1 May 2022). GSEA was performed and visualized with the clusterprofiler [19] R-package, and the perturbation was set at 1000. Gene sets with Normalized Enrichment Score (NES) > 1.0 and False Discovery Rate (FDR) < 0.25 could be deemed as those with significant enrichment of TPM4-related genes.

### 2.5. Gene Set Variation Analysis (GSVA)

To identify the potential signaling pathways through which TPM4 was involved in the epithelial–mesenchymal transition (EMT) process, GSVA was performed. Seven canonical EMT-related signaling pathways, concluded by Gonzalez et al. [20], were obtained from the GSEA website. GSVA analysis was conducted with the GSVA package and visualized with the Corrgram package.

### 2.6. Single-Cell Sequencing Analysis

Sc-RNAseq analysis was performed with the Seurat package. After data normalization, a total of 2000 highly variable genes were identified via FindVariableGenes function. Subsequent principal component analysis (PCA) was performed with RunPCA, followed by FindNeighbors and FindClusters to cluster cells with a resolution of 0.1. Finally, results were presented by the UMAP method, and cell markers were utilized for cell annotation [21]. The expression level of TPM4 and cell markers were visualized with VlnPlot and Dimplot across different cell clusters. Single-cell pseudotime trajectory analyses on tumor cells and macrophages were performed with the Monocle package according to the standard workflow [22]. Briefly speaking, cells were dimensionality-reduced and arranged into a trajectory. As a result, tumor cells were divided into seven states with three branch points, and macrophages were separated into three states with one branch point.

### 2.7. Statistical Analysis

R language (R Foundation), with a range of packages (pheatmap, pROC, ggplot2, GSVA, circlize, corrgram, and survival), was utilized to conduct the statistical analyses. Gaussian distribution was performed before data analysis. Kaplan–Meier (KM) curves were established to evaluate the differences between TPM4-low and TPM4-high groups across different WHO grades. Cox proportional hazard regression model was conducted using coxph function provided in the survival package and visualized with forestmodel function. All statistical tests were two-sided. A *p* value of <0.05 was defined as statistically significant.

## 3. Results

### 3.1. TPM4 Was Associated with More Malignant Characteristics of Gliomas

In both CGGA and TCGA datasets, comparisons of TPM4 levels revealed significant positive correlations between WHO grade and TPM4 expression (Figure 1A,E). With further subclassification of IDH mutation status, we found that IDH wildtype was more associated with upregulation of TPM4, though the statistical difference was not significant in some subgroups (Figure 1B,F), which also exhibited apparent trends. When taking into account the molecular subtype, TPM4 expression was significantly upregulated in the mesenchymal subtype (Figure 1C,G), which usually predicts a much worse survival status in glioma. These results enlightened us that TPM4 might contribute as a potential predictor for mesenchymal gliomas. Figure 1D,H show the receiver-operating characteristic (ROC) curves, which demonstrate the area under the ROC curve (AUC) was 81.7% in TCGA and 87.8% in CGGA.

### 3.2. TPM4-Related Biological Function

The potential molecular function of TPM4 for gliomas was investigated based on TPM4-high-correlated genes. We performed Pearson correlation tests to search for TPM4-high-correlated genes based on Pearson coefficients. Through filter criteria of correlation coefficients > 0.6, 664 TPM4 positively correlated genes from the CGGA dataset and 855 from the TCGA dataset were screened out. An intersection of TPM4 significantly correlated genes between CGGA and TCGA was subsequently chosen for GO analysis. Finally, 344 overlapping genes (Appendix A) were found to be consistently correlated with TPM4 in both datasets and were then annotated by GO analysis. As shown in Figure 2A,B, genes that highly correlated with TPM4 mainly participated in the regulation of extracellular matrix/structure organization, response to wounding, and vasculature development, as well as cell adhesion.

Given that GBM is a distinctive group of gliomas, additional GO analyses were further performed for GBM. A total of 95 overlapping TPM4-correlated genes were screened out (Appendix A), and they were shown to be mainly correlated with biological adhesion, cell adhesion, cell motion, and extracellular matrix/structure organization, which were similar to those in pan-gliomas (Figure 2C,D).

### 3.3. TPM4 Was Tightly Associated with EMT

GSEA was conducted in each dataset further to demonstrate the biological function of TPM4 in gliomas. TPM4 expression showed the most significant correlation with the EMT phenotype in both datasets (CGGA NES = 1.95, FDR = 0.04 and TCGA NES = 1.94, FDR = 0.06) (Figure 3A,B,E,F). Moreover, a similar pattern was consistently observed among GBM in both datasets (Figure 3C,D,G,H). These results enlightened us that TPM4 might be involved in the EMT process during gliomagenesis.

### 3.4. Relationship between TPM4 and EMT-Pathways

The correlation between TPM4 and EMT signaling pathways was further investigated. Seven genesets (Appendix A), representing different types of pro-EMT pathways, concluded by Gonzalez [20], were obtained on the GSEA website and were performed with cluster analysis according to TPM4 expression. As Figure 4A,B show, three genesets consisting of hypoxia, TGF-β, and PI3K/AKT showed significant correlations with TPM4 in both CGGA and TCGA datasets. To validate the results of cluster analysis, the seven pro-EMT signaling pathways were transformed into corresponding metagenes via Gene Set Variation Analysis (GSVA). The interactive relationship between TPM4 and seven metagenes was demonstrated through Corrgrams (Figure 4C,D). There was a significant association between TPM4 and hypoxia, PI3K/AKT, as well as TGF-β signaling pathway, respectively, consistent with that we observed in cluster analyses. In contrast, TPM4 exhibited a relatively weak relationship with other signaling pathways. Furthermore, taking GBM as a distinct group, the association between TPM4 and pro-EMT signaling pathways was further investigated in GBM. Results showed that other than hypoxia PI3K/AKT, and TGF-β, TPM4 concurrently revealed a remarkable relationship with MAPK (Appendix A).

### 3.5. TPM4 Was Associated with Key Biomarkers of EMT

In order to further explore the potential role of TPM4 in the EMT phenotype of gliomas, we evaluated the co-expression association between TPM4 and EMT biomarkers. Through Pearson correlation and Circos plots (Figure 5A,B), TPM4 was found to be significantly correlated with mesenchymal biomarkers (N-cadherin, Snail, and Slug). In addition, for the GBM subgroup, we also observed a strong correlation between TPM4 and these EMT biomarkers (Figure 5C,D). These results indicated that TPM4 might interact with these molecules in the EMT process of glioma. Moreover, TPM4 showed a very weak relationship with E-cadherin, which may be attributed to that E-cadherin was more regulated at protein level rather than transcriptome level. Several other biomarkers are defined as the pivotal molecules in EMT [23], such as TWIST1, TWIST2, β-catenin, and vimentin. We additionally performed a correlation analysis between these biomarkers and TPM4, and it turned out that TPM4 expression showed a tight association with vimentin and TWIST1 (Appendix A), which were also typically mesenchymal markers.

### 3.6. Abnormal TPM4 Expression in Single-Cell RNAseq Data

To identify the cell types that highly express TPM4, CGGA sc-RNAseq data were analyzed. Eight cell types were clustered and visualized with the UMAP method (Figure 6A). Based on the cell markers (Appendix A), cell clusters were annotated (Figure 6B). Clusters 0, 1, 3, and 5 overexpressing PDGFRA and EGFR were annotated as tumor cells. Clusters 2 and 6 highly expressing CD68 and C1QC were annotated as the monocyte-macrophage linage, and cluster 4 overexpressing MOG was concluded as oligodendrocytes. Cluster 7, highly expressing CD3D and CD3E, represented a series of T cells. As shown in Figure 6C,D, TPM4 could be expressed by tumor cells, macrophages, oligodendrocytes, and T cells in the tumor microenvironment (TME). It is worth noting that TPM4 expression exhibited an upregulating trend in tumor cells and macrophages as the WHO grade increased, in contrast to the uniform distribution of TPM4 expression among oligodendrocytes and T cells across different WHO grades (Figure 6E,F).

To explore the TPM4 expression status during distinct cell developmental stages of tumor cells and macrophages, a subsequent single-cell trajectory analysis was performed on tumor cells and macrophages. In tumor cells, seven cell states were displayed (Figure 6G). As revealed in the pseudotime analysis (Figure 6H), state 1 was defined as the early stage of tumor initiation, and states 4, 5, and 6 could be deemed as developmental or established tumor cells. It turned out that TPM4 was mainly expressed and activated in developmental or established tumor cells (Figure 6I and Appendix A), especially in state 5 of tumor cells. Moreover, in macrophages, three cell states were identified (Figure 6J). Pseudotime trajectory (Figure 6K) revealed that state 1 was defined as the early-stage or naïve macrophages, and states 2 and 3 could be deemed as mature or polarized macrophages. TPM4 expression was mainly expressed by mature macrophages and correlated with macrophage M1/M2 polarization (Figure 6L and Appendix A). Totally, these results suggested that the abnormal expression level of TPM4 (As the disease progresses, TPM4 increases) in glioma was mainly attributed to the increased expression of TPM4 in tumor cells and macrophages and associated with tumor progression and macrophage polarization.

### 3.7. Higher Level of TPM4 Predicted Unfavorable Outcome in Gliomas

Kaplan–Meier (KM) curve was generated to investigate the prognostic value of TPM4 in both datasets. According to the median value of TPM4 expression, pan-glioma patients in both datasets were divided into two groups. We observed that higher TPM4 was associated with worse overall survival than their counterparts (Figure 7A,D). Furthermore, similar patterns of the KM curves were concurrently revealed among LGG (Figure 7B,E) and GBM patients (Figure 7C,F), respectively. To investigate the independent prognostic value of TPM4, univariate and multivariate Cox regression models were performed in both cohorts, and the results showed that upregulation of TPM4 could independently predict the unfavorable outcome in the CGGA dataset (HR = 1.81, *p* = 0.001) (Figure 7G), while in TCGA multivariate Cox model, because of collinearity between TPM4 and IDH mutation status, the prognostic value of TPM4 was not significant (HR = 1.49, *p* = 0.058) (Figure 7H). Considering the ethnicity heterogeneity of the TCGA population, we also repeated the Cox analysis restricting data to Caucasians and found that TPM4 appeared to be a determinant of poor prognosis independently within the Caucasian subgroup (HR = 1.28, *p* = 0.036) (Figure 7I). However, because of the small numbers of non-Caucasian participants in TCGA, our results were limited to the Caucasian population. Altogether, these results indicated that TPM4 could be identified as an effective prognosticator among Chinese and Caucasian patients with gliomas.

## 4. Discussion

Emerging evidence indicates that TPM4 is a crucial modulator during tumor development. TPM4 has been reported as an oncogene across different malignancies, including oral squamous cell carcinoma [3], breast cancer [4,5], lung cancer [6,7], hepatocellular carcinoma [8], and ovarian cancer [9], while in uterine cervix cancer [10] and colon cancer [11], higher TPM4 indicated a better prognosis. However, the clinical significance and expression profile of TPM4 in glioma remains unclear.

This study comprehensively analyzed the expression patterns of TPM4 at the transcriptional level and its clinical characterization in whole WHO grade of gliomas based on 998 glioma patients. To explore the TPM4 expression status in all gliomas, we took advantage of the CGGA dataset, including mRNA microarray data of whole-grade glioma. To further validate what we have revealed in the CGGA dataset, we obtained and analyzed RNAseq data of glioma from the TCGA network, and we found that the consistency of results between the two cohorts was overall satisfying. Our results revealed a significant relationship between TPM4 upregulation and more malignant phenotype in gliomas, including GBM, mesenchymal molecular-subtype, and IDH-wildtype glioma. TCGA molecular subtype and IDH mutation status have been well-validated as both prognostic and predictive markers in gliomas. The mesenchymal-subtype glioma, one of the TCGA molecular subtypes, which expresses mesenchymal biomarkers, has the worst prognosis of the subtypes [24]. IDH-wildtype gliomas are more aggressive, difficult to resect, and insensitive to chemotherapy, particularly temozolomide, resulting in shorter survival times than are found in IDH-mutant gliomas [25]. Altogether, these results demonstrated that higher TPM4 expression was paralleled with more malignant clinical and pathophysiological characteristics in gliomas. Moreover, upregulation of TPM4 predicted significantly worse survival for glioma patients across different WHO grades. These results indicated a potential pro-tumoral function of TPM4 in gliomas, consistent with its effect in most other malignant tumors reported previously. Unveiling the biological function of TPM4 in glioma cells may facilitate a better understanding of the molecular mechanisms of gliomagenesis.

Through GO analysis, TPM4 significantly correlated genes were found to be highly correlated with extracellular-structure organization, extracellular-matrix organization, cell adhesion, and biological adhesion, which are vital steps of cell migration and invasion. This suggested that TPM4 and the TPM4-correlated genes might interact and play a synergistic effect in the remodeling of extracellular matrix, cell migration, and invasion of gliomas, in line with previous studies regarding other types of cancer. TPM4 was reported to promote tumor invasion, migration, and metastasis in lung cancer [6], hepatocellular carcinoma [8], and breast cancer [4]. Furthermore, GSEA analysis revealed a robust relationship between TPM4 and EMT phenotype, which had been identified as a vital process in tumor migration, invasion, recurrence, and therapeutic resistance in glioma cells [26,27,28]. Thus, TPM4 was considered to be potentially involved in the EMT development in gliomas.

To further demonstrate the relationship between TPM4 and EMT phenotype, a panel of pro-EMT signaling pathways and key biomarkers was subsequently included in the analysis to demonstrate their interactions with TPM4. The results revealed a significant association between TPM4 and PI3K/AKT, hypoxia, and TGF-β signaling pathways. Additionally, TPM4 displayed strong positive correlations with the majority of the mesenchymal biomarkers, including Snail, Slug, N-cadherin, TWIST1, and vimentin. These results suggested that TPM4 may profoundly interact with these pro-EMT signaling pathways and biomarkers, further demonstrating the profound participation of TPM4 during the EMT process. 

The reliability of the single-cell RNAseq technique for exploring the biological characteristics of particular genes has been widely reported [29,30]. Our scRNA-seq analysis demonstrated that TPM4 was ubiquitously expressed on neoplastic cells, macrophages, oligodendrocytes, and T cells in glioma TME. In tumor cells and macrophages, TPM4 expression showed an increasing trend as the WHO grade increased, suggesting that TPM4 abnormality was mainly attributed to tumor progression and tumor-associated macrophage (TAM) infiltration, which was further confirmed in the subsequent pseudotime trajectory analysis. As a critical cell subpopulation of TME, TAMs have been concluded to play a vital role in promoting the genesis and progression of glioma [31,32]. Particularly, M2 polarization of macrophages is significantly associated with aggressive progression across different malignancies, especially with immune escape of tumor cells. Moreover, a substantial body of research has formed a growing recognition of the interplay between TAMs and EMT regulation. TAMs could enhance the EMT process through multiple molecules and signaling axes, such as CCL2 [33], IFN-γ [34], IL-1β, IL-8, TNF-α, and TGF-β [35]. These results further validated the essential role of TPM4 in regulating the EMT process in glioma progression.

While our study was being completed, a similar conclusion about the potential role of whole TPM family members in gliomas was reached in another report, which was an impressive study presented by Huang et al. [36]. Through a systematic analysis of the TPM family based on the expression profile in the TCGA dataset, they identified that TPM3 might play an oncogenic role and contribute as a novel biomarker for poor prognosis in gliomas. Unique to our analysis, through an integrative study of the molecular characteristics, prognostic value, as well as the biological function of TPM4 expression deriving from TCGA and CGGA datasets, we highlight that TPM4 is another critical predictor of poor prognosis and might correlate with malignant biological behaviors during gliomagenesis, which would further deepen our understanding of the potential role of the TPM family in glioma.

## 5. Conclusions

In conclusion, TPM4 is closely related to more aggressive characteristics of glioma, and TPM4 upregulation predicts a shorter survival time for glioma patients. Although the exact mechanisms of TPM4 warrant further research, this study demonstrates that TPM4 plays a vital role during the pro-EMT process via its synergistic interactions with pro-EMT signaling pathways and key molecules. However, our analysis has its limitation. No biological experiments were performed, and future molecular biological studies are required to validate the function of TPM4 in gliomas.

## Figures and Tables

**Figure 1 brainsci-12-01120-f001:**
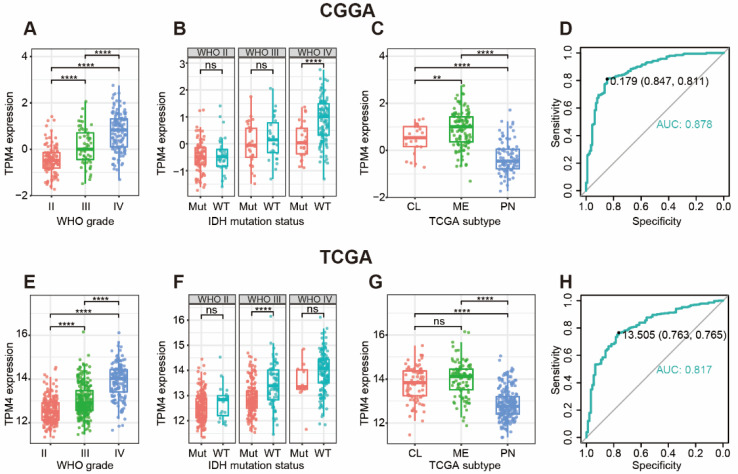
TPM4 expression according to WHO grade (**A**,**E**), IDH mutation status (**B**,**F**), TCGA molecular subtype (**C**,**G**), and receiver-operating characteristic (ROC) curves (**D**,**H**) for differentiating mesenchymal subtype. ** indicates *p* value < 0.01, **** indicates *p* value < 0.0001. ns: not significant; Mut: IDH mutation; WT: IDH wildtype; CL: classical subtype; ME: mesenchymal subtype; PN: proneural subtype; AUC: area under ROC curve.

**Figure 2 brainsci-12-01120-f002:**
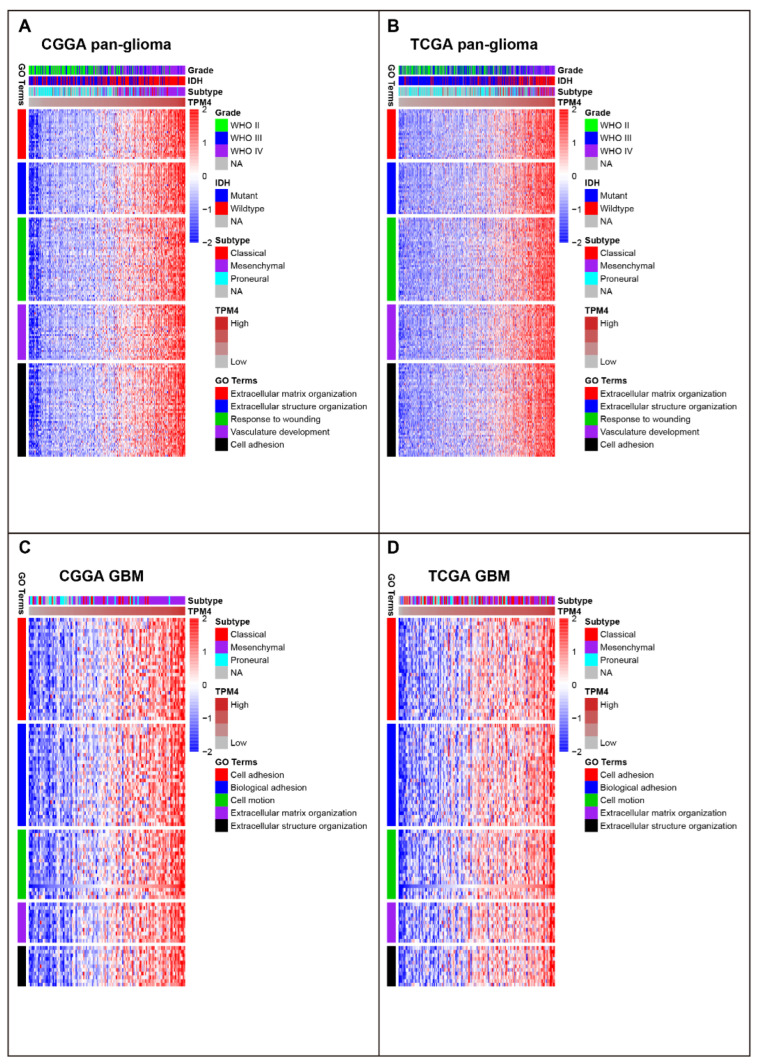
Gene Ontology (GO) analysis for TPM4 in pan-glioma (**A**,**B**) and GBM (**C**,**D**). Statistics value of pan-glioma GO: extracellular matrix organization (gene count = 25, *p* = 2.57 × 10^−18^), extracellular structure organization (gene count = 26, *p* = 1.44 × 10^−14^), response to wounding (gene count = 42, *p* = 1.47 × 10^−12^), vasculature development (gene count = 28, *p* = 7.76 × 10^−12^), and cell adhesion (gene count = 47, *p* = 1.75 × 10^−11^). Statistics value of GBM GO: cell adhesion (gene count = 28, *p* = 6.82 × 10^−16^), biological adhesion (gene count = 28, *p* = 7.03 × 10^−16^), cell motion (gene count = 19, *p* = 1.45 × 10^−10^), extracellular matrix organization (gene count = 11, *p* = 3.78 × 10^−10^), and extracellular structure organization (gene count = 11, *p* = 3.04 × 10^−8^). NA: not available.

**Figure 3 brainsci-12-01120-f003:**
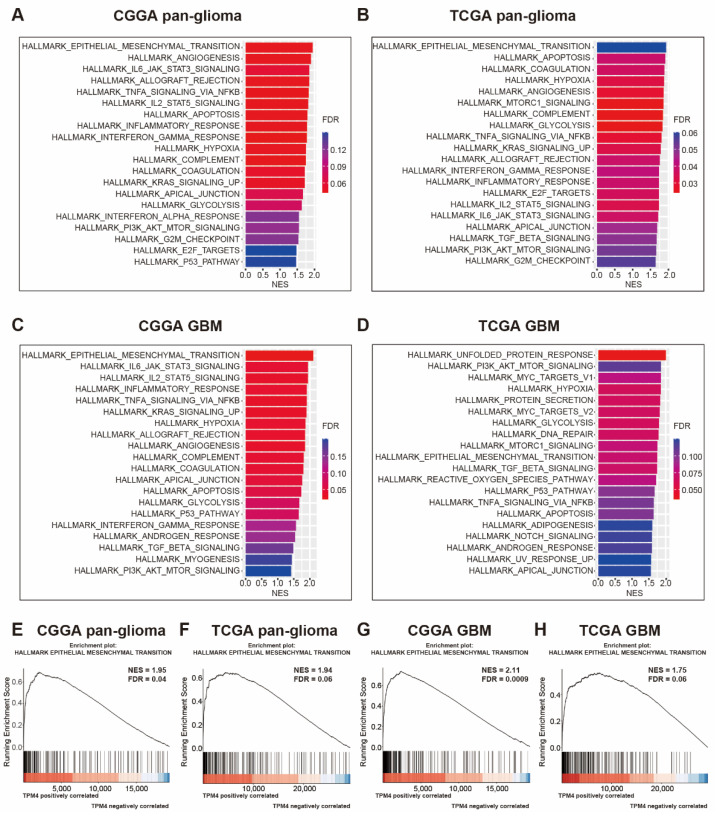
Gene set enrichment analysis (GSEA) for TPM4 in pan-glioma (**A**,**B**) and GBM (**C**,**D**), and GSEA plots for enrichment of epithelial-to-mesenchymal transition according to TPM4 expression in pan-glioma (**E**,**F**) and GBM (**G**,**H**). NES: normalized enrichment score; FDR: false discovery rate.

**Figure 4 brainsci-12-01120-f004:**
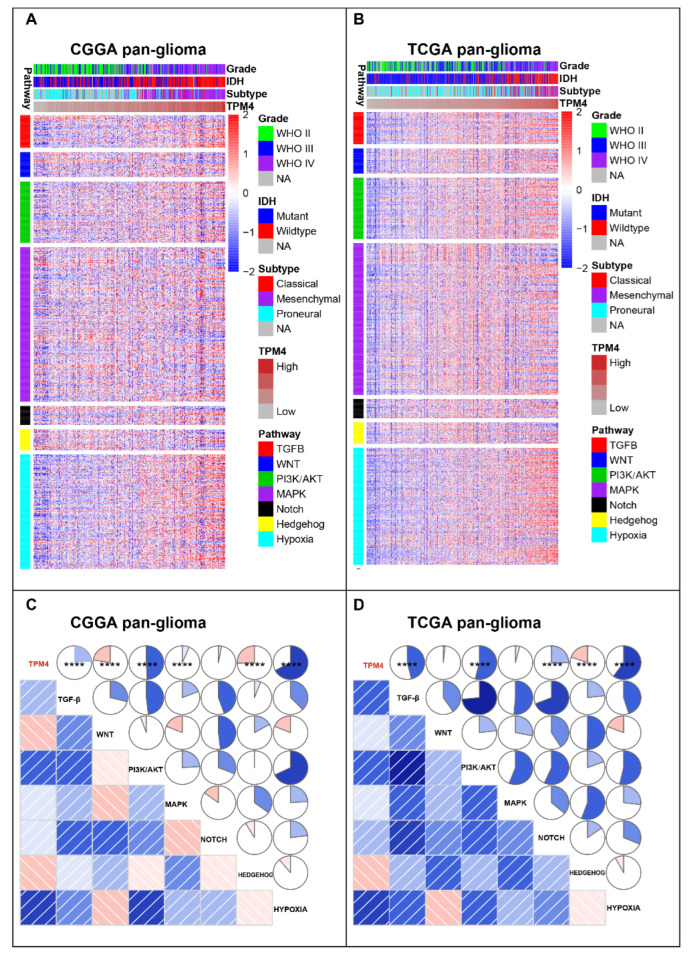
Cluster and gene set variation analysis (GSVA) of TPM4-related epithelial-to-mesenchymal transition (EMT) signaling pathways in pan-glioma. (**A**) Heatmap of representative genes of different EMT-related pathways in CGGA pan-glioma. (**B**) Heatmap of representative genes from different EMT-related pathways in TCGA pan-glioma. (**C**) Intercorrelation between TPM4 and seven metagenes in CGGA pan-glioma. (**D**) Intercorrelation between TPM4 and seven metagenes in TCGA pan-glioma. In (**C**,**D**), the blue and red colors represent positive and negative correlation, respectively. A darker color and a bigger sectorial area represent a higher correlation coefficient. **** indicates *p* value < 0.0001. In CGGA, Pearson correlation between TPM4 and *TGF-*β (R = 0.240, *p* = 2.542 × 10^−5^), WNT (R = −0.225, *p* = 7.919 × 10^−5^), PI3K/AKT (R = 0.493, *p* < 2.2 × 10^−16^), MAPK (R = 0.065, *p* = 0.264), NOTCH (R = 0.029, *p* = 0.616), HEDGEHOG (R = −0.256, *p* = 6.805 × 10^−6^), HYPOXIA (R = 0.672, *p* < 2.2 × 10^−16^). In TCGA, Pearson correlation between TPM4 and TGF-β (R = 0.461, *p* < 2.2 × 10^−16^), WNT (R = 0.017, *p* = 0.661), PI3K/AKT (R = 0.538, *p* < 2.2 × 10^−16^), MAPK (R = 0.040, *p* = 0.309), NOTCH (R = 0.252, *p* = 7.126 × 10^−11^), HEDGEHOG (R = −0.197, *p* = 4.083 × 10^−7^), HYPOXIA (R = 0.599, *p* < 2.2 × 10^−16^); NA: not available.

**Figure 5 brainsci-12-01120-f005:**
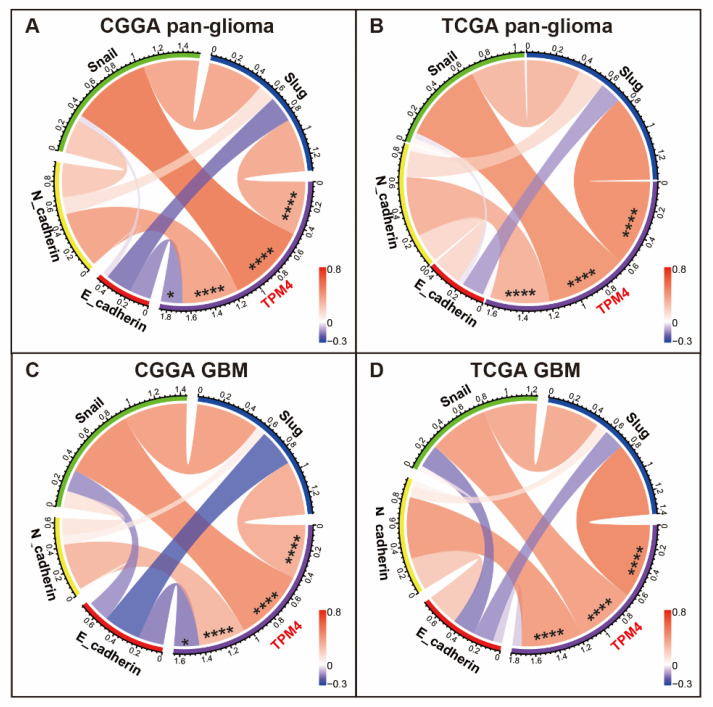
Correlation analysis between TPM4 and key EMT biomarkers in pan-glioma (**A**,**B**) and GBM (**C**,**D**). (**A**) Correlation analysis in CGGA pan-glioma. (**B**) Correlation analysis in TCGA pan-glioma. (**C**) Correlation analysis in CGGA GBM. (**D**) Correlation analysis in TCGA GBM. * indicates *p* value < 0.05. **** indicates *p* value < 0.0001.

**Figure 6 brainsci-12-01120-f006:**
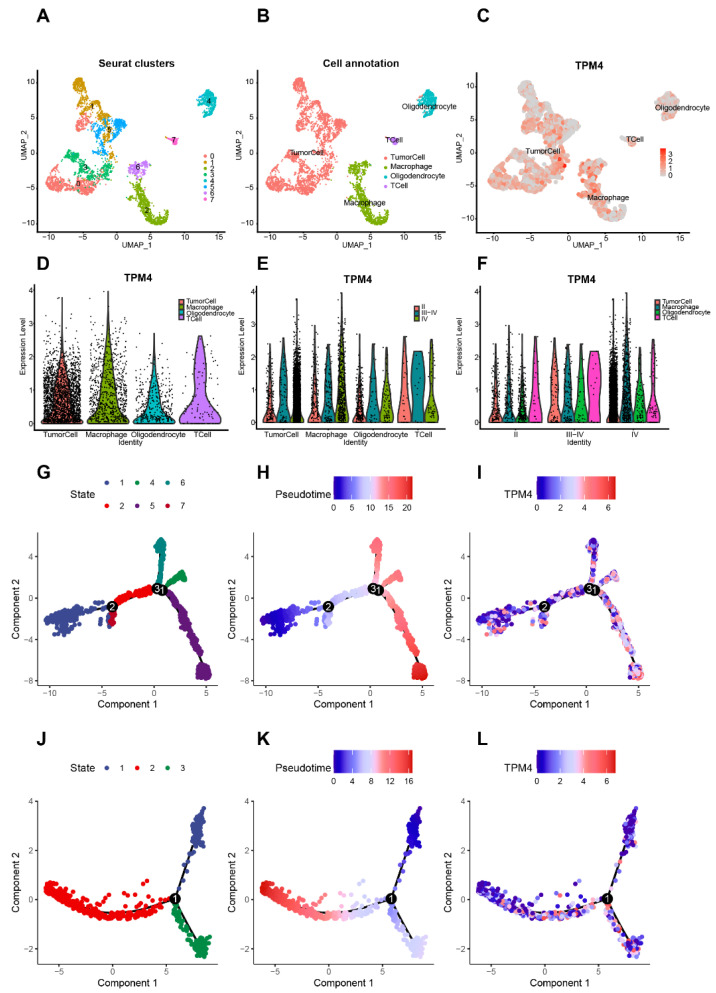
TPM4 expression on different cell types based on CGGA scRNA-seq data. (**A**) Eight cell types were clustered with Seurat package. (**B**) Cell annotation according to cell markers (Appendix A). Clusters 0, 1, 3, and 5 overexpressing PDGFRA and EGFR were annotated as tumor cells. Clusters 2 and 6 highly expressing CD68 and C1QC were annotated as the monocyte-macrophage linage, and cluster 4 overexpressing MOG was concluded as oligodendrocytes. Cluster 7, highly expressing CD3D and CD3E, represented a series of T cells. (**C**) Dimplot of TPM4 expression across different cell types. (**D**) Vlnplot of TPM4 expression across different cell types. (**E**,**F**) TPM4 expression across different subgroups. (**G**–**I**) Single-cell developmental trajectory analysis of tumor cells revealed seven states. Cells were colored based on states (**G**), pseudotime (**H**), and TPM4 expression (**I**). (**J**–**L**) Single-cell developmental trajectory analysis of macrophages. Cells were colored based on states (**J**), pseudotime (**K**), and TPM4 expression (**L**). In pseudotime developmental trajectory analysis, the black circled numbers indicates different branch points.

**Figure 7 brainsci-12-01120-f007:**
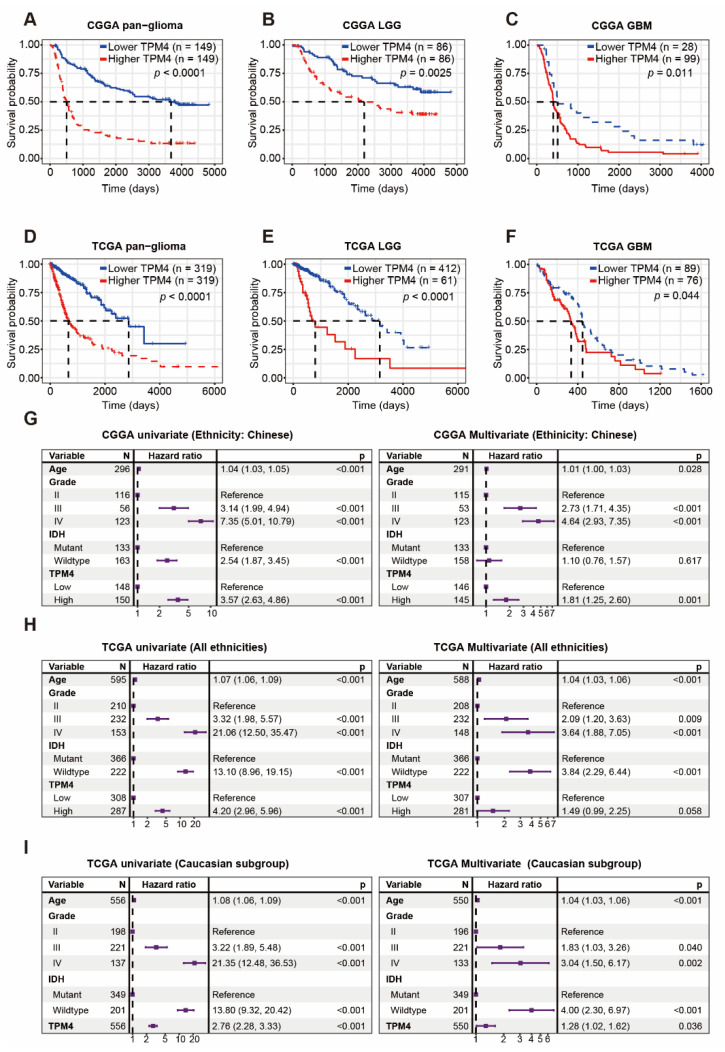
Prognostic value of TPM4 in glioma. (**A**–**C**) Kaplan–Meier survival analysis according to TPM4 in CGGA microarray dataset. (**D**–**F**) Kaplan–Meier survival analysis according to TPM4 in TCGA RNAseq dataset. (**G**) Univariate and multivariate Cox regression model in CGGA microarray dataset (ethnicity: Chinese). (**H**) Univariate and multivariate Cox regression model in TCGA RNAseq dataset (all ethnicities). (**I**) Univariate and multivariate Cox regression model in TCGA RNAseq dataset (Caucasian subgroup).

## Data Availability

The datasets supporting the conclusions of this article are available from The Cancer Genome Atlas (TCGA) dataset (http://cancergenome.nih.gov/, accessed on 10 June 2020) and the Chinese Glioma Genome Atlas (CGGA) dataset (http://www.cgga.org.cn/, accessed on 26 July 2021).

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
