# Peer review of "Clinical Characterization and Prognostic Value of TPM4 and Its Correlation with Epithelial–Mesenchymal Transition in Glioma"

_brainsci, 2022, doi:10.3390/brainsci12091120_

Round 1

Reviewer 1 Report

In this manuscript the authors evaluate the clinical and prognostic role of TPM4 in gliomas. While the purpose of the manuscript is of interest it is important to consider recent literature with a similar content and conclusions (The Tropomyosin Family as Novel Biomarkers in Relation to Poor Prognosis in Glioma, Ke Huang et al 2022, Biology). Additionally, I find that the methodology is very poorly described often obscuring how did the authors reach their conclusions. The graphical presentation would also gain from a better explanation in the legends. Overall figures are often repetitive and some data is redundant for a main figure and could be considered to be transferred to supplementary or combined (see comments below). Some statements in the main text are not supported by the results.

Major comments

1)     Over the whole manuscript there is often an inconsistency between the CGGA and TCGA datasets. For instance in Fig 1 B and F, Fig 3 C and D, Fig 4 C and D, Fig 5 C and D, Fig 7 as well as in Fig 9. Could the authors comment on these differences and explain also why datasets where processed separately. Of particular interest is the differences between CGGA and TCGA in the multivariate analysis for TPM4, which is not significant in the CGGA. This makes one consider if some differences are specific to some ethnic groups and authors should include this variable in this analysis.

2)     As mentioned above a recent article by Ke Huang et al has looked in the role of TPM4 in glioma, can the authors indicate what are the new findings of their article compared to this one? It is important to include it as a reference as well.

3)     Please indicate a more detailed explanation on all methodology. In particular, pseudotime trajectory, cell types annotations (where these done by you or used the ones from the datasets?), how where different cell states defined? How were grades defined for the scRNA seq analysis?  Pearson correlation test for TPM4 what p value was used? In Figure 3 it is not clear how TPM4 was correlated to EMT.

4)     Some statements are not supported by the results or are unclear. In particular line 139, line 151 (I do not see the differences to which the authors refer in the figure); line 165 (I also do not see this result in the figures); line 167 (This is a bold statement, E-cadherin might be regulated on protein level and not on RNA seq). Statements related to pseudotime results are not well supported by the data. Please show violin or boxplot of expression of TPM4 in the different states found with pseudotime. Line 204: this statement is contradictory to Line 188.

5)     Fig 4 and 5 C and D are not clearly explained in the legend. What are the pie charts? What does color scale indicate. In general check the figures to make sure that they are more comprehensive.

6)     Throught the text often results are stated as different between groups but statistics to support it are missing (e.g Line 189; 198; 202; Fig 4-5 (C and D))

Minor comments:

1)     In the legends and throughout the text it is often unclear what is represented. Indicate what is KM, ROC, AUC

2)     Line 50: the protein atlas states that TPM4 is not a prognostic factor for glioma. Please comment on this. Include the adequate references. https://www.proteinatlas.org/ENSG00000167460-TPM4/pathology/glioma

3)     Fig 1 indicate in the legend what is different between panels. What is ROC and AUC?

4)     Line 165, Snail and Slug should be with capital letter.

5)     A lot of figures are redundant and could be moved to the supplementary data or combined into one (eg Fig 6 and 7 which even have exactly the same legend; Fig 4 and 5)

Reviewer 2 Report

Glioblastoma is the most common type of brain cancer in adults and can interrogate through brain tissue to make it hard to remove completely. Despite the combined therapeutic approach of surgery, radiation, and chemotherapy, the patients’ life is expected to be short within 15-18 months. The key failure of the therapeutic approach is due to the resistance of radiotherapy, chemotherapy, or the targeted drugs. By using the CGGA and TCGA dataset, Wang et al., found that TPM4 was associated with epithelial to mesenchymal transition (EMT). High TPM4 level was observed in tumor cells associated with unfavorable outcomes; therefore, TPM4 protein level could be used as a marker to predict poor prognosis in patients with glioma.

The article is well written and informative. The results found in this manuscript are only performed by computational and biological experiments to confirm their findings. It could be another project for their future work.

I only have several minor comments:

  1. Please use Table S1, S2 instead of Table SI, SII.

  2. What is the author's conclusion about the expression level of TPM4 in Classical, Messenchymal, and Proneural? I did not see any discussion about the differences in these subtypes.

  3. What is the author's conclusion about the expression level of TPM4 in IDH mutant and wildtype? Please discuss in the discussion part.

  4. What does NA (Figure 2, 4) and ZZ (Figure 5) mean? Please define them in the Figure legends.

  5. There is a cut off in Figure 8D.

Author Response

Comments and Suggestions for Authors
Glioblastoma is the most common type of brain cancer in adults and can interrogate through brain tissue to make it hard to remove completely. Despite the combined therapeutic approach of surgery, radiation, and chemotherapy, the patients' life is expected to be short within 15-18 months. The key failure of the therapeutic approach is due to the resistance of radiotherapy, chemotherapy, or the targeted drugs. By using the CGGA and TCGA dataset, Wang et al., found that TPM4 was associated with epithelial to mesenchymal transition (EMT). High TPM4 level was observed in tumor cells associated with unfavorable outcomes; therefore, TPM4 protein level could be used as a marker to predict poor prognosis in patients with glioma.

1. The article is well written and informative. The results found in this manuscript are only performed by computational and biological experiments to confirm their findings. It could be another project for their future work.

Response: Dear professor, thanks very much for your careful review. We really appreciate your professionalism in this area. And we are very grateful for your valuable suggestions. According to your kind comments and suggestions, we now have revised the manuscript. I sincerely wish that you could kindly agree. Thanks a lot for your valuable comments and precious time. With best regards.

I only have several minor comments:

1.Please use Table S1, S2 instead of Table SI, SII.

Response: Dear professor, thanks very much. We appreciate your carefulness in reviewing our manuscript. Now we have revised the words accordingly. Tables SI, SII, SIII, and SIV have been replaced
with Tables S1-4. Thanks a lot for your work.

2.What is the author's conclusion about the expression level of TPM4 in Classical, Messenchymal, and Proneural? I did not see any discussion about the differences in these subtypes.

Response: Dear professor, thanks very much for your careful review. It is our great honor to have your valuable suggestions. Now we have added the contents in the Discussion Section to demonstrate the significance of TPM4 difference across different TCGA molecular subtypes. Revisions are listed as follows. Thanks very much.

TCGA molecular subtype and IDH mutation status have been well-validated as both prognostic and predictive markers in gliomas. The mesenchymal-subtype glioma, one of the TCGA molecular subtypes,
which expresses mesenchymal biomarkers, has the worst prognosis of the subtypes[1]. IDH-wildtype gliomas are more aggressive, difficult to resect, and insensitive to chemotherapy, particularly temozolomide, resulting in shorter survival times than are found in IDH-mutant gliomas[2]. Altogether, these results demonstrated that higher TPM4 expression was paralleled with more malignant clinical and pathophysiological characteristics in gliomas.

3.What is the author's conclusion about the expression level of TPM4 in IDH mutant and wildtype? Please discuss in the discussion part.

Response: Dear professor, thanks very much for your careful review. It is our great honor to receive your review comments. We have revised in the Discussion Section as follows, which we hope that you could kindly agree with. Thanks for your precious time.

TCGA molecular subtype and IDH mutation status have been well validated as both prognostic and predictive markers in gliomas. The mesenchymal-subtype glioma, one of the TCGA molecular subtypes,
which expresses mesenchymal biomarkers, has the worst prognosis of the subtypes[1]. IDH-wildtype gliomas are more aggressive, difficult to resect, and insensitive to chemotherapy, particularly temozolomide, resulting in shorter survival times than are found in IDH-mutant gliomas[2]. Altogether, these results demonstrated that higher TPM4 expression was paralleled with more malignant clinical and pathophysiological characteristics in gliomas.

4.What does NA (Figure 2, 4) and ZZ (Figure 5) mean? Please define them in the Figure legends.

Response: Dear professor, thanks very much for your careful review. We are very sorry for our unclear description. It’s our great honor to have your meticulous comments. NA indicates not available. ZZ was a typo error. During the analysis procedure, NA was a special data type rather than a character type in R language software. If NA was directly used, the NA could not be displayed in the final heatmaps. So when we performed corresponding analyses, we substituted the NA with ZZ. When we submitted our manuscript, the incorrect version of the figure was uploaded. Now we have replaced the new figure. We  feel really sorry for this. Furthermore, Professor Reviewer 1 thinks that Figure 5 might be redundant and should be placed in the supplementary files. Now, Figure 5 has been put into the supplementary Figure S1, which we sincerely hope that you could kindly agree. Thanks very much for your careful review and precious time.

5.There is a cut off in Figure 8D.

Response: Dear professor, thanks very much for your valuable suggestions. We really appreciate your professionalism in this field. In Figure 8D, what we would like to express was that TPM4 could be expressed by tumor cells, macrophages, oligodendrocytes, and T cells in the tumor microenvironment (TME). We do understand that you kindly request us to reveal the cutoff value for distinguishing the tumor cells and macrophages from other types of cells. When we attempted to perform receiver operating characteristic (ROC) analysis, we found that there were a large number of zero values in the sc-RNAseq data; it seemed to be very challenging to conduct ROC analysis for such a kind of data. Furthermore, we have reviewed several previous studies regarding sc-RNAseq data[3-5]; similarly, cutoff values for identifying different cell types were also unavailable in these high-impact studies. Thus, we cordially wish that you could kindly agree with the current form of the figure panel (Figure 8D in the original submission version, while Figure 6D in the revised manuscript). Thanks very much for your precious time and meticulous review. Best regards.

References

1.
Pan, Y. B.; Zhu, Y.; Zhang, Q. W.; Zhang, C. H.; Shao, A.; Zhang, J., Prognostic and Predictive Value of a Long Non-coding RNA Signature in Glioma: A lncRNA Expression Analysis. Front Oncol 2020, 10, 1057.
2.
Choi, K. S.; Choi, S. H.; Jeong, B., Prediction of IDH genotype in gliomas with dynamic susceptibility contrast perfusion MR imaging using an explainable recurrent neural network. Neuro Oncol 2019, 21 (9), 1197-1209.
3.
Yu, K.; Hu, Y.; Wu, F.; Guo, Q.; Qian, Z.; Hu, W.; Chen, J.; Wang, K.; Fan, X.; Wu, X.; Rasko, J. E.; Fan, X.; Iavarone, A.; Jiang, T.; Tang, F.; Su, X. D., Surveying brain tumor heterogeneity by single-cell RNA-sequencing of multi-sector biopsies. National Science Review 2020, 7 (8), 1306-1318.
4.
Williams, D. W.; Greenwell-Wild, T.; Brenchley, L.; Dutzan, N.; Overmiller, A.; Sawaya, A. P.; Webb, S.; Martin, D.; Genomics, N. N.; Computational Biology, C.; Hajishengallis, G.; Divaris, K.; Morasso, M.; Haniffa, M.; Moutsopoulos, N. M., Human oral mucosa cell atlas reveals
a stromal-neutrophil axis regulating tissue immunity. Cell 2021, 184 (15), 4090-4104 e15.

5.
Chen, R.; Wang, X.; Dai, Z.; Wang, Z.; Wu, W.; Hu, Z.; Zhang, X.; Liu, Z.; Zhang, H.; Cheng, Q., TNFSF13 Is a Novel Onco-Inflammatory Marker and Correlates With Immune Infiltration in Gliomas. Front Immunol 2021, 12, 713757.

Round 2

Reviewer 1 Report

Thanks for the revision provided, however I feel that there are a few points that can still be improved or that were not sufficiently adressed.  (See my previous numbering for the comments)

Major comments

1) Please include the information about the ethnicity composition of each dataset, I feel that this is an important point that should not be missed by the readers.

2) The paragraph about the work of Ke Huang is written in a very colloquial language, please re-write to make it more formal.

4) Please provide the information that you give in the reponse to the comment also in the legend of the figure. I am still missing a colorcode that would be absolute. It is only stated what blue and red means (negative or positive correlation), but it is not clear how much is it really correlated. I would also provide here the p values of the correlation (perhaps on top of the pie charts). 

For the pseudotime, the violin or boxplot that I suggested previously can also be performed outside of Monocle scope. The RNA assay and state (Fig 6 G and J) information can be extracted out of the Monocle object and ploted with ggplot. I think this would be more supporting of the results since I still can not grasp the difference in fig 6 I and L.

5) please expand more in the legend

6) As stated above, would be helfful to view the p values on top of the pie chart of the correlation plot.
